# Temporal Range Dynamics of the Lataste’s Viper (*Vipera latastei* Boscá, 1878) in Doñana (Spain): Insights into Anthropogenically Driven Factors

**DOI:** 10.3390/ani14203025

**Published:** 2024-10-19

**Authors:** Rafael Carmona-González, Francisco Carro, Juan Pablo González de la Vega, Fernando Martínez-Freiría

**Affiliations:** 1Estación Biológica de Doñana (EBD-CSIC), Avda. Américo Vespucio, 45, 41092 Seville, Spain; pcarro@ebd.csic.es; 2Atlas Herpetológico de Andalucía, Avda. Andalucía, 70. 4°D, 21007 Huelva, Spain; latasti@hotmail.com; 3CIBIO, Centro de Investigação em Biodiversidade e Recursos Genéticos, InBIO Laboratório Associado, Universidade do Porto, 4485-661 Vairão, Portugal; 4BIOPOLIS Program in Genomics, Biodiversity and Land Planning, CIBIO, 4485-661 Vairão, Portugal

**Keywords:** ecological niche models, niche shift, climate change, landscape changes, conservation, Iberian Peninsula, Viperidae

## Abstract

Doñana (Spain) is a region of notable biodiversity richness, threatened by landscape transformation and climate change. Lataste’s viper (*Vipera latastei*), a snake endemic to the Iberian Peninsula, has one of the few remaining coastal populations in Doñana, which is likely to be declining. Our study shows that the viper’s suitable habitat has been shrinking over time toward the coastal area. By analyzing data from the past 50 years, we found that rising temperatures and habitat changes have led to a sharp decline in suitable environments for the species. These findings underscore the urgent need for targeted conservation measures to protect this threatened snake.

## 1. Introduction

Anthropogenic factors such as landscape transformation and climate change are threatening global biodiversity [1,2,3]. Significant alterations to natural habitats have been taking place in the past century, leading to the emergence of contemporary anthropogenic biomes [4]. These alterations still persist or have even increased due to the continuous demand for resources by humans [5]. Yet contemporary research has highlighted the importance of climate change in the ongoing biodiversity decline [6,7]. Both anthropogenic factors are synergistically interacting, producing changes in geographical distribution, phenology, and local abundance of species [8,9], therefore increasing their risk of extinction [10,11]. 

Ecological niche-based models (ENMs) constitute a major tool to investigate species distributions and the factors driving them [12,13]. ENMs correlate species occurrence data with environmental variables to identify the realized ecological niche of species, also enabling predictions of the species potential distribution over time [13,14]. ENMs have been widely used in climate change research to forecast future species distributions under climate change scenarios and assess their level of exposure to the ongoing trends in temperature and precipitation (e.g., [15,16]). Other studies have used ENMs to compare occurrence data from distinct periods, drawing inferences about the historical range dynamics of species and identifying potential shifts in the realized niche due to the influence of anthropogenic factors (e.g., [17,18]). Undoubtedly, ENM-based approaches have the potential to guide the development of conservation strategies, particularly for endangered species in protected areas [18,19].

The Mediterranean Basin is one of the world’s biodiversity hotspots, harboring high levels of biodiversity and endemism, both of which are threatened by human activities [20,21]. Within it, Doñana (Huelva, southern Spain) is a region of notable biodiversity richness, inhabited, for example, by some 400 species of birds, 50 species of mammals, and 25 species of reptiles (https://www.miteco.gob.es/es/parques-nacionales-oapn/red-parques-nacionales/parques-nacionales/donana/visita-virtual/fauna.html, (accessed on 30 September 2024)). It is recognized through distinct establishments and levels of protection (e.g., both national and natural parks, UNESCO Biosphere Reserve [22]). Similarly to other regions in the Mediterranean Basin, Doñana has been traditionally affected by human activities, but, in the last decades, these activities have increased, producing discernible effects on the provision of ecosystem services and remarkably affecting the local landscape [23]. This transformation has led to the current isolation of the region, fundamentally altering its ecological dynamics and functions [24]. Recently, Doñana has been under the spotlight due to the illegal withdrawal of groundwater for intensive agriculture purposes [25]. In addition to these changes in land use, Doñana is also facing the challenges of climate change: recent assessments have shown a decrease in rainfall and an increase in temperatures [26,27]. All of these factors, therefore, are pushing the region toward environmental collapse. 

Anthropogenic landscape and climate change are major threats to reptile conservation worldwide [28]. Dependence on environmental conditions (i.e., ectothermic physiology), combined with specific biological traits (e.g., low dispersal, diet specialization, low reproductive output), increase populations vulnerability to these anthropogenic factors, making vipers (Fam. Viperidae) among the most threatened reptiles worldwide [29]. In this regard, the Lataste’s viper, *Vipera latastei* Boscá, 1878, is considered the most endangered snake in the Iberian Peninsula [30], as well as one of the most endangered snakes in the Mediterranean Basin, listed as Vulnerable by the IUCN Red List [31]. Phylogenetically closely related to the North African *Vipera monticola*, the Lataste’s viper is a medium-sized venomous snake endemic to the Iberian Peninsula, where it inhabits areas with a Mediterranean climate, occupying humid, sub-humid, and semi-arid climatic sub-types [32,33,34,35]. Along its range, *V. latastei* mostly occurs in scattered and isolated populations restricted to mountain ranges, showing moderate-to-low population densities, especially in the south of its distribution range [36,37,38]. Its distribution area was likely more extensive in the past, but, currently, populations are fragmented and isolated due to anthropogenic activities [36,37,38,39]. In addition, the species is predicted to be severely affected by climate change, with extensive reductions in its current potential distribution, ranging from 74% to 78% by the period of 2041–2070 [15].

Apart from general studies (e.g., the description of *V. latastei gaditana* with type specimens from Doñana [40]), limited biological research has focused on the population of the Lataste’s viper in the region. Worth mentioning is the study by José Antonio Valverde demonstrating the structure of the vertebrate community in Doñana and providing valuable information on the biology and status of the viper species [41]. According to that study, the Lataste’s viper in Doñana is common in the marshlands, *Halimium* scrublands, and coastal dunes [41]. Currently, the species seems absent from marshland habitats, being overall considered uncommon by many researchers and herpetologists. Indeed, most of the records of the Lataste’s viper in Doñana come from roadkills [42] and online platform accounts listing scarce contemporaneous records for the species in this region (e.g., SIARE—Servidor de Información de Anfibios y Reptiles de España. n.d. Available online: https://siare.herpetologica.es/bdh/distribucion, accessed on 7 October 2023). The population of the Lataste’s viper in Doñana represents one of the few remaining coastal populations of the species in the whole Iberian Peninsula [35]. It seems to be isolated by both natural barriers (e.g., rivers Tinto-Odiel and Guadalquivir) and human infrastructure (e.g., highway, agricultural areas) from the closest population (i.e., Sierra Morena, SIARE—Servidor de Información de Anfibios y Reptiles de España. n.d. Available online: https://siare.herpetologica.es/bdh/distribucion, accessed on 7 October 2023). Therefore, ecological research on the population of the Lataste’s viper in Doñana is crucial to understand the species resilience to anthropogenic factors, overall contributing to its conservation.

In this study, we compile records of the Lataste’s viper in Doñana to investigate the effect of landscape transformation and climate change on the species distribution over time. For this purpose, we rely on an ENM approach that allows us to identify and compare the realized ecological niche of the species in two time periods (i.e., historical and contemporary periods). We expect a decrease in the range size of the species in current times, due to the influence of anthropogenic changes in the landscape and/or climate. We also expect to recover some signals regarding the capacity of the species to shift its realized niche and cope with these anthropogenic changes. Our specific aim is to answer the following questions: (1) “which environmental factors are most related to the distribution of the species in historical and contemporary times in Doñana?”, (2) “which are the suitable areas for species occurrence in historical and contemporary times?”, and, (3) “how has the extension of suitable areas for the species changed over time?”. By answering these questions, we also expect to contribute to the improvement of the ecological knowledge of the species in the region and assist its conservation.

## 2. Materials and Methods

### 2.1. Study Area

Doñana is located in the Huelva province, southern Spain. Our study area was delimited as a squared area (4464 km^2^) based on the distribution of the Lataste’s viper population in Doñana (see next section), such a population being spatially restricted by the estuaries of both the Odiel-Piedras River and the Guadalquivir River and by the southern part of the A-49 highway (Figure 1).

The regional climate exhibits characteristics of a sub-humid Mediterranean climate pattern, influenced by the Atlantic Ocean. The annual mean temperature hovers around 17 °C. The coldest month, typically corresponding to January, generally registers a mean temperature of approximately 10 °C, while the warmest month, usually July, experiences a mean temperature of around 24 °C. Mean annual precipitation is around 550 mm, although it exhibits significant year-to-year variability, ranging from a low of 170 mm during the 2004–2005 period to a high of 1000 mm in 1995–1996 [43]. Rainfall primarily occurs between October and April, with a distinct dry season spanning from May to September [44]. Due to this precipitation regime, the water balance is generally negative except for 3–4 months of the year [45].

Three main biotopes can be distinguished in the study area: (1) marshlands on a clay area with an irregular flood pattern during winter, (2) mobile sand dunes parallel to the seacoast, and (3) scrubland that, according to the proximity to the water table, is either xerophytic Mediterranean scrubland (drier) or hygrophytic Atlantic scrubland [46]. Reforestation of *Pinus pinea* has been carried out over the last three centuries [47], altering scrublands and dunes.

The study area includes several areas of protection such as the National Park, Natural Park, UNESCO Biosphere Reserve, Ramsar Site, and World Heritage Site [22]. It contains the largest wetland in western Europe, an intricate matrix of marshes (270 km^2^), phreatic lagoons, a 25 km long dune ecosystem with its respective coastline, and representative Mediterranean terrestrial plant communities.

### 2.2. Species Records

We compiled a total of 415 records for the period of 1959–2022, for which coordinates up to 1 × 1 km^2^ resolution could be gathered (Figure 1). Records were collected from (a) the live and roadkill specimens found during field work conducted by the authors and collaborators (see acknowledgements) for “Atlas Herpetológico de la provincia de Huelva” and other scientific purposes (*n* = 349), (b) the animals collected for scientific purposes (live or roadkill) that are currently available in the Estación Biológica de Doñana (EBD) scientific collection (*n* = 41), (c) field notebooks of the EBD rangers and other investigators (*n* = 20), and (d) citizen science platforms such as GBIF (https://www.gbif.org/, accessed on 29 September 2022), iNaturalist (https://www.inaturalist.org/, accessed on 29 September 2022), and Observation (https://observation.org/, accessed on 29 September 2022) (*n* = 5). Records with no specific date (*n* = 23) were removed for subsequent analyses. The examination of the number of records based on 10-year periods clearly shows an increase in the records until the decade of 1990 and a descent since the 2000s (Figure 1). Therefore, we established the year 2000 as a threshold to separate historical and contemporary times.

For ecological modeling purposes, records were first assigned to the corresponding 1 × 1 km UTM grid (European Datum 1950 Zone 29) and then to a specific time period (i.e., until 1999 and from 2000 to 2022). Duplicated records (i.e., same 1 × 1 km UTM grid for the same period) were removed. Consequently, of 249 records collected until 1999 and 188 collected between 2000 and 2022, only 86 and 102, respectively, were used for ecological modeling. Note that, although there are more records available for the period of 2000–2022 than for the period of 1959–1999, the range they represent is smaller than the range recovered for the period of 1959–1999 (797.093 km^2^ vs. 1298.855 km^2^, as measured by minimum convex polygons; see Appendix A). For the purpose of this study, contemporary records were assigned to the recent past period (for a similar procedure see [17,18]), and, therefore, two datasets were created: (1) the historical dataset (*n* = 190), comprising all records from 1959 to 2022, and (2) the contemporary dataset (*n* = 112), comprising records for the period of 2000–2022 only. 

Species records were obtained from uneven sampling and could, therefore, bias predictions from the ecological models [13,48]. To mitigate this potential bias, we spatially rarefied the records for both time periods using the SDMtoolbox 2.0 in ArcMap 10.5 [49]. In brief, both datasets were spatially rarefied by taking into account the distance among them (three classes; from 1 km to 3 km) to remove records within clusters, while considering the environmental variability of the study area (as described by the environmental variables used to run ecological models for each period, see below) to avoid removing records from areas with a heterogenous climate [49]. The final datasets included 49 and 47 observations for the historical and the contemporary times, respectively (see Appendix A).

### 2.3. Climatic Factors

Climatic variables represent important predictors of the environmental suitability with respect to ectotherms (e.g., [34,50,51]). To perform our ecological models, we relied on a set of 19 bioclimatic variables that are biologically meaningful and are derived from the monthly temperature and rainfall values (see [52]). We obtained monthly values of maximum and minimum temperature, as well as precipitation data for the period from 1980 to 2020 at a spatial resolution of 30 arc-seconds (approximately 1 km^2^ at the equator) from the CHELSA online repository (https://chelsa-climate.org/, accessed on 20 January 2023). We further averaged the data for the historical (1980–1999) and the contemporary (2000–2020) periods. Monthly average values for these time periods were calculated using the ‘raster’ package in R 4.2. Subsequently, we created 19 bioclimatic variables (Appendix A) for each period using the ‘dismo’ package in R 4.2 and the ‘biovars’ function. These bioclimatic variables were then imported into ArcGIS 10.5 and projected to European Datum 1950 zone 29. Following this, the variables were clipped to the study area and the pixel size was transformed into 1 × 1 km.

### 2.4. Normalized Difference Vegetation Index (NDVI) Factors

The normalized difference vegetation index (hereafter NDVI) is a widely used metric for quantifying the presence of live green vegetation using sensor data [53,54]. NDVI values span from −1 to +1, with negative values indicating the absence of vegetation [54]. NDVI was obtained from “Servidor de Imágenes Landsat y productos derivados de Doñana” [43] at a 1 × 1 km UTM grid (European Datum 1950 Zone 29) for the historical (1984–1999) and contemporary (2000–2020) periods. Worth mentioning is that NDVI data from before 1984 were not available. For each period, we first averaged the rasters by month and then we calculated the mean and standard deviation values for three seasonal periods, spring (March to May), summer (July and August), and autumn (September to November). Consequently, we obtained six variables for each period as initial predictors of our ecological models (Appendix A). The data were obtained. The NDVI was calculated using the raster calculator and cell statistics tools in ArcGIS v 10.5.

### 2.5. Ecological Niche-Based Models

The nature of our observations (i.e., coming from different sources) and the biological characteristics of the vipers (i.e., low detection probability) precluded the use of ecological models based on presence–absence methods. For these reasons, ecological models were developed using the maximum entropy approach on the presence–background software Maxent 3.4.3 [55]. This modeling technique has been reported to deliver a strong performance in many ecological modeling studies, even with a low sample size [56,57], and has been extensively used in ecological studies on *V. latastei* and other viper species at distinct scales (e.g., [34,50,58,59]).

Two models were developed for each period, i.e., historical (before the year 2000) and contemporary (after the year 2000), using associated variables from the corresponding time. To avoid collinearity in the predictors used to train the ecological models [13], spatial correlation among all variables was tested (i.e., the 19 bioclimatic + 6 NDVI for each period), and only nine weakly correlated variables (R < 0.75) were retained for each period (Table 1; Appendix A). The correlation matrix of the variables for each period was obtained using the ‘modEvA’ package in R [60].

In order to produce ecologically robust models (see [48]), we used the ‘ENMeval’ R package [61] to assess model parameterization (i.e., features and regularization parameter) before running the final models. Consequently, both historical and contemporary models were run with linear features and a regularization parameter of 3, as these parameters showed the highest performance (see Appendix A). In each model, we ran a total of 20 model replicates with a random seed, partitioning the occurrence data in 80%/20% to train/test each replicate (i.e., 40/9 in the historical models, 39/9 in the contemporary models). Occurrences for each replicate were chosen by bootstrap, allowing for sampling with replacement. The area under the curve (AUC) of the receiver operating characteristics (ROC) plot was taken as a measure of the models’ performance [62]. In addition to the model predictions for each period (i.e., historical and contemporary), the historical model was projected into the contemporary period (referred to as ‘projection’). This approach allowed us to predict the habitat suitability in contemporary times, considering the ecological relationships that were found in the historical times, thus addressing a potential niche shift over time (see [17,18]).

The weight of the variables for describing the species distribution was determined by its average percentage contribution and permutation importance to the models. The relation between occurrence of vipers and variables was determined through visual examination of response curve profiles from univariate models (e.g., [17,50]).

To convert models of continuous suitability into binary predictions (i.e., absence–presence), we used two commonly implemented thresholds, the minimum training (e.g., [63]) and the 10th percentile logistic (e.g., [64]) thresholds. The first threshold delimits the proportion of test localities with suitability values lower than that associated with the lowest-ranking training locality, while the second indicates the proportion of test localities with suitability values lower than that which would be obtained by excluding the 10% of training localities with the lowest predicted suitability; omission rates greater than the expectation of 10% typically indicate model overfitting [65].

In the GIS software, binary historical, contemporary, and projection data were compared to quantify the temporal dynamics of available suitable areas in the absence/presence of a potential shift in the realized niche. Consequently, we calculated the following categories: (1) lost, i.e., pixels identified in historical times only, representing the suitable areas that were lost in contemporary times, (2) underestimated, i.e., pixels identified in the contemporary and historical data but not in the projection data, referring to the pixels where the species could currently occur, reflecting a niche shift, (3) permanent, i.e., pixels identified in historical, projection, and contemporary data, referring to the pixels where the species could currently occur, without considering a niche shift, and (4) gained, i.e., pixels identified in the contemporary data only, representing pixels that became suitable in contemporary times, reflecting a niche shift.

## 3. Results

### 3.1. Models’ Performance

The overall quality of both historical and contemporary models was deemed to be satisfactory, as they exhibited high levels of performance in both training (avg ± SD AUC historical = 0.865 ± 0.024; avg ± SD AUC contemporary = 0.9 ± 0.021) and testing datasets (avg ± SD AUC historical = 0.842 ± 0.07; avg ± SD AUC contemporary = 0.879 ± 0.057). 

### 3.2. Environmental Factors Related to Species Occurrence

In both the historical and contemporary models, the mean temperature of the driest quarter (TempDri) was recognized as the most important variable affecting species distribution (by both percentage of contribution and permutation importance metrics; Table 2 and Appendix A). In the historical model, species distribution was also affected by the mean temperature of the wettest quarter (TempWet) and by the standard deviation of NDVI in spring (Spring SD). However, these variables exerted a minimum effect on species distribution in the contemporary models (Table 2 and Appendix A).

Response curve profiles for the most important variable in both historical and contemporary models, i.e., the mean temperature of the driest quarter (TempDri), showed a similar response, with species occurrence in both time periods mostly occurring at lower temperature values (Figure 2, Appendix A). However, the response curve of the contemporary period w displaced in relation to the response curve of the historical period.

### 3.3. Predicted Occurrence

In both models, areas of high suitability for the species occurrence were mostly restricted to the coastal region (Figure 3). These areas were predicted to be larger in extent in the historical model in comparison to the contemporary model and more fragmented in the projection model in comparison to the contemporary model (Figure 3).

Considering the minimum training logistic threshold, the binary historical model predicted 1696 km^2^ (37.99% of the study area) to be suitable for the species, while the contemporary model predicted a smaller area compared to the historical model as suitable for the species (1179 km^2^, 26.41% of the study area) (Figure 4). In addition, the projection of the historical model into the contemporary conditions predicted 282 km^2^ (6.32% of the area) as suitable (Figure 3).

### 3.4. Range Dynamics

When comparing the historical model and its projection into contemporary times (i.e., the temporal dynamics in the absence of a niche shift), 282 km^2^ were identified as permanently suitable, while 1414 km^2^ lost their suitability over time (Figure 4). Therefore, predicted suitable areas within the study area were found to experience an 83.37% decrease from the historical to the projection period. These declines occurred in most of the historical suitable habitats, in the north, central, and east of the study area.

Comparing the projected and the contemporary period (i.e., measuring shifts in the realized niche), 282 km^2^ were identified as permanently suitable, while 897 km^2^ were underestimated as suitable habitats, i.e., unsuitable pixels in the projected data that were classified as suitable in the contemporary data (Figure 4). Therefore, considering shifts in the realized niche, predicted suitable areas within the study area were found to experience a decrease of 30.5% from historical to contemporary times. These declines occurred in the central and east regions of the study area.

Model and projection predictions considering the 10th percentile logistic threshold provided similar quantifications in relation to species temporal dynamics and realized niche (see Appendix A).

## 4. Discussion

This study relies on a thorough compilation of occurrence records and on an ecological modeling approach to identify and quantify the environmental factors related to the Lataste’s viper distribution in Doñana. Our data describe a pattern of species rarefication over time, supporting a likely reduction in its distribution range from historical (before the year 2000) to contemporary times (after the year 2000). The execution of this study across two distinct time frames (historical and contemporary) allowed the estimation of the factors behind such a range reduction, contributing to the assessment of how the species is being affected by the ongoing anthropogenic changes. In addition, our results indicate further lines of research that may contribute to the conservation of the species in the region.

### 4.1. Temporal Rarefication of the Lataste’s Viper in Doñana

In their natural environments, snakes frequently show low population densities and engage in extended periods of inactivity, which importantly reduce observations [66]. Most species possess elusive and secretive tendencies, and vipers are no different in this regard. Indeed, *V. latastei* is a species characterized by a striking secretive and elusive behavior, making it exceptionally challenging to obtain precise demographic, behavioral, and genetic data through conventional field research methods [38,39,67]. 

Our data show that *V. latastei* is mainly restricted to the coastal area of Doñana and that its occurrence becomes rarer over the more than 50 years considered in this study. Although our study is not based on a systematic sampling approach, it considers an exhaustive and extensive dataset which includes the sampling efforts made by distinct researchers and the EBD monitoring team to cover the area of the national and natural parks. Specifically, we consider the records available from the EBD coming from the specimens stored in the scientific collection and from the field notebooks of Doñana rangers produced since 1965. In addition, our dataset comprises the records gathered in the “Atlas Herpetológico de la provincia de Huelva”, which reflects the sampling efforts in the adjacent areas of Doñana over more than 40 years. Therefore, we are confident of the reliability of our dataset and the described pattern of restriction to coastal areas and rarefication over time found for *V. latastei* in Doñana.

### 4.2. A Distribution Driven by Cool Temperatures

Our results indicate that one climatic factor, the mean temperature of the driest quarter (TempDri), majorly explains the distribution of *V. latastei* in Doñana (Table 2). Another climatic factor (the mean temperature of wettest quarter, TempWet) and one NDVI factor (Spring SD) play a role in the species’ distribution (in the historical model only), although emerging as less important than the mean temperature of the driest quarter (Table 2). It is important to acknowledge that our study was conducted at the local scale in a specific region, and, therefore, our ENM outputs describe the local niche of the species, being this just a part of what the species could express across its distribution range (see [68]). Nonetheless, climatic factors have also been found to be the primary drivers of the distribution of other reptiles [69,70], as well as of the distribution of *V. latastei* at both global/regional [34,71] and local scales [50,68,72].

Remarkably, a previous study conducted at the scale of the Iberian Peninsula reported that the topographic factors ‘slope’ and ‘altitude’ had emerged as the most significant factors associated with the viper’s distribution [39]. Far from being an incongruity in relation to our study and the abovementioned studies, the study by Santos et al. (2006) [39] highlighted what national atlases [36,37] and other studies conducted at the regional scale (e.g., [34,37]) can attain: the identification of the greatest restriction of the species to the mountain ranges, where the coldest environments of the Mediterranean Iberia are found. In our study, we found a pattern of species restriction to cool environments when examining the response curve profiles of the mean temperature of the driest quarter (Figure 2). The proximity to the Atlantic Ocean provides areas with cool conditions, which allow the species to thrive in Doñana. We speculate that the historical extirpation of coastal populations of the Lataste’s viper (besides Doñana, only a few populations currently persist, as for instance: Vila do Conde, Porto, northern Portugal and Cabo de Gata, Almería, southeastern Spain [35,38]) could have biased our understanding of the ecological requirements of the species in mountain ranges. Increasing knowledge of coastal populations of *V. latastei* is, therefore, key to have a better picture of the ecology of the species and determine the best conservation management approaches.

### 4.3. A Reduction in the Distribution Range Driven by Climate Change

Our study identifies changes in the environmental correlates and potential distribution of the Lataste’s viper in Doñana over time. We found that the species distribution was majorly driven by three variables in the historical period, while only one variable importantly relates to the species distribution in contemporary times (Table 2). The realized niche of the species in the historical period was more complex than the realized niche of the species in the contemporary time, a phenomenon which suggests a reduction in the niche breadth over time (see [17,68,73]). At the same time, we found a reduction in the potential distribution range of the species from historical to contemporary times (see historical vs. contemporary, Figure 4). A contraction of the realized niche is likely to be a common phenomenon in species decline, being usually linked to a reduction in species distribution caused by the intensity of threats and/or the capacity to tolerate threats varying across the niche space [17,73]. 

One climatic variable, the mean temperature of the driest quarter (TempDri), was the most important in the models developed for both time periods. Although response curve profiles for this variable show a similar pattern in both the historical and the contemporary models, there is a notable displacement in the profiles which translated to an increase in the selected temperatures by the viper from historical to contemporary times (Figure 2). Interestingly, a detailed look at the range of these variables in both time periods gives us the clue for understanding this shift in terms of climatic preferences: there was an increase in the range of this variable from historical to current times (0.4 °C in the minimum value, Table 1), meaning that the minimum values of this temperature variable in the historical period are no longer available in Doñana. Climate change is one of the major threats to Lataste’s viper conservation, as it is expected to reduce the potential range of the species in the coming years [15]. Our results are consistent with the possible effects of climate change on the species’ distribution gathered at the regional scale. In Doñana, it must be considered that significant reductions in annual precipitation and increases in minimum temperatures are expected by 2060 [74]. Our results, therefore, warn about the vulnerability of *V. latastei* in the upcoming years and about the potential reductions that can occur at the local scale in this region. 

We found out that *V. latastei* may have some potential capacity to cope with the ongoing climate change. In absence of a niche shift, the species would be more restricted to the coastal area as shown by the potential range of our projection of the historical model into contemporary times (see projection vs. contemporary, Figure 4). Usually ignored in correlative climate change assessments (e.g., [15,58]), local adaptation and phenotypic plasticity are major mechanisms allowing species to cope with the ongoing climate change [75]. For instance, behavioral changes can be developed to accommodate the increase in temperatures, ultimately affecting the activity and phenology of individuals and populations [76,77]. Accordingly, many contemporaneous observations of *V. latastei* in Doñana have occurred at night, while diurnal observations, even in the early morning and late afternoon (corresponding to daily peaks of activity in the species, [38]), are scarce (authors, personal observations). These observations would suggest a shift toward nocturnal activity in *V. latastei* that would allow it to cope with the current increase in temperatures. Nevertheless, this hypothesis requires to be addressed through other techniques such as biophysical modeling [78] or radiotracking monitoring (e.g., [79]).

A recent study using remote sensing and ENMs has shown that European and Iberian reptiles are highly threatened by landscape transformation [80]. Landscape transformation, with an increase in intensive agriculture in current times, was one of the expected factors likely driving this process of range reduction in *V. latastei* in Doñana. Although *V. latastei* can occupy a variety of agricultural habitats (e.g., in north-central Spain [81]), it is generally absent from areas where intensive farming occurs, since it totally transforms shelter and resources on which this snake depends [38]. In Doñana, agriculture intensification has occurred in the north and western regions, also affecting the groundwater level and leading to the desiccation of several lagoons and the aridification of surrounding areas [25]. Meeting our expectation, the lack of importance of the variable NDVI spring SD in current times could be reflecting such a landscape transformation. However, there is not a remarkable spatial variation in this variable from historical to current times (Appendix A) and, thus, our study does not uncover an effect of agriculture intensification on the species’ range contraction. While we believe that this could be due to an effect of the time frame of our study (i.e., we considered NDVI images up to 2020, overseeing the most recent years of drought and aridification [25,82]), we encourage the development of alternative remote sensing studies that aim to monitor ongoing landscape transformation of major habitats in Doñana (e.g., [83]).

## 5. Conclusions

Doñana represents one of the last remaining coastal populations of the Lataste’s viper. Understanding the ecology of this population is key in a species for which the few local scale ecological information that exists come from mountainous regions (e.g., [79,84]). Our results show that the species’ distribution is primarily restricted to areas characterized by cool temperatures, near the coastal area, and that there is a concerning trend whereby the species’ range could be increasingly restricted. This phenomenon is here attributed to the increase in temperatures favored by the ongoing climate change, mirroring the situation faced by many mountainous species, which are experiencing a reduction in range size at the same time in which they track their ecological requirements, increasing in elevation [85,86]. 

Our study warns about the long-term viability of this coastal population, particularly when the specific biological traits of this species (e.g., low dispersal, diet specialization, low reproductive output) are considered. Nevertheless, further research should be conducted to better assess the conservation of this population and anticipate management actions. Among others, we believe that the following two major research lines should be a priority: (1) the establishment of a monitoring program within the protected areas to gain insights into the ecological, demographic, activity, and reproductive traits of the species in this region [67], and, importantly, to anticipate the potential effects of both landscape and climate changes [87], also including sea level rise, and (2) the quantification of the genetic diversity of the population of *V. latastei* in Doñana, also determining its level of isolation in relation to neighboring populations, as a way of preventing any potential genetic erosion (e.g., [88,89,90]).

## Figures and Tables

**Figure 1 animals-14-03025-f001:**
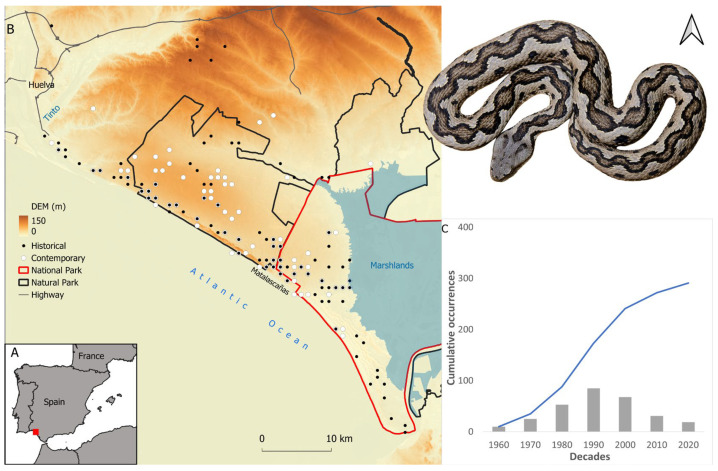
(**A**) Location of the study area in the south west of the Iberian Peninsula. (**B**) The study area depicting the distribution of the records considered for the historical (black dots) and contemporary (white dots) times, Doñana national and natural parks, highways, altitude, and marshlands. (**C**) Histogram depicting the total and accumulative number of records per decade. A female Lataste’s viper from Doñana is included in the figure (Author: G. Martínez del Mármol).

**Figure 2 animals-14-03025-f002:**
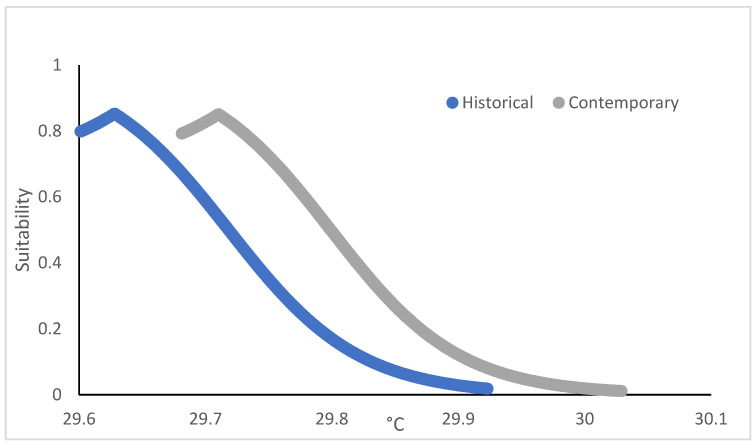
Response curve profiles for the mean temperature of the driest quarter, the most important bioclimatic factor related to the historical and contemporary distributions of the Lataste’s viper in Doñana.

**Figure 3 animals-14-03025-f003:**
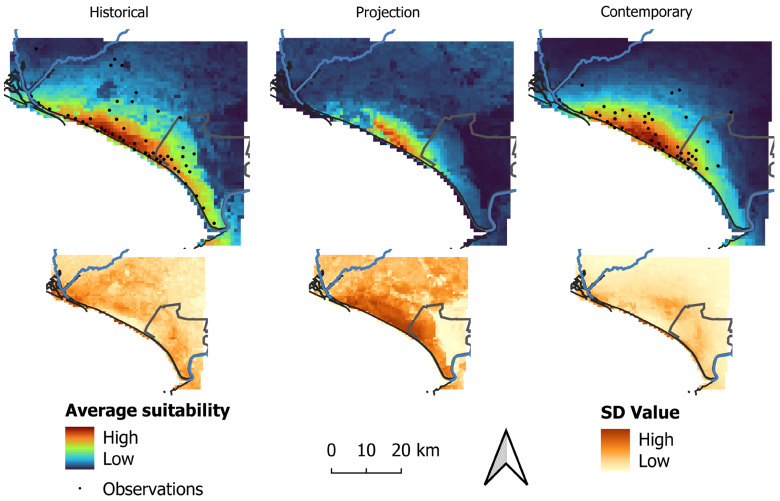
Average (**top**) and standard deviation (SD, **bottom**) for habitat suitability of the Lataste’s viper in the historical, contemporary, and projection models.

**Figure 4 animals-14-03025-f004:**
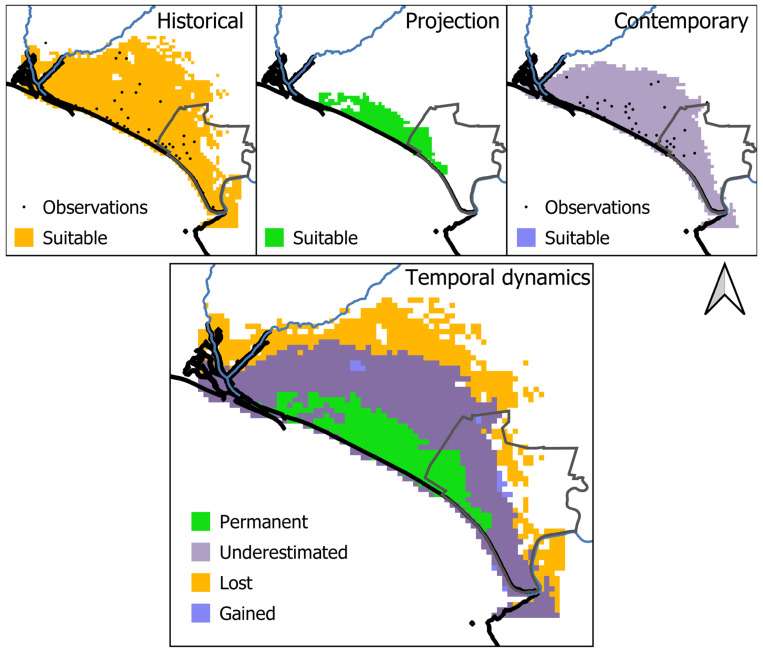
On the top, the historical, contemporary, and projected representation of suitable habitats for the Lataste’s viper in Doñana. On the bottom, temporal dynamics of habitat suitability for the Lataste’s viper in Doñana: (i) lost habitats represent the suitable areas that were lost in contemporary times, (ii) permanent suitable habitats refer to pixels where the species could currently occur, without considering a niche shift, (iii) underestimated suitable habitats refer to pixels where the species could currently occur, mediated by a niche shift, and (iv) gained suitable habitats refer to pixels that became suitable in contemporary times, reflecting a niche shift.

**Table 1 animals-14-03025-t001:** Bioclimatic and NDVI variables considered for ecological niche modeling purposes, depicting code, meaning (and units), and ranges of variation for the historical and contemporary periods.

Code	Meaning (and Units)	Historical	Contemporary
AnnTemp	Annual mean temperature (°C)	29.079–29.192	29.118–29.226
TempWet	Mean temperature of wettest quarter (°C)	28.515–28.705	28.803–28.967
TempDri	Mean temperature of driest quarter (°C)	29.628–29.896	29.71–30
PrecDri	Precipitation of driest month (mm)	3.751–1.274	0.881–0.498
PrecSeas	Precipitation seasonality (coefficient of variation)	72.966–77.987	66.263–70.08
Autumn SD	Autumn standard deviation	0.03–0.238	0.026–0.271
Spring M	Spring mean	−0.499–0.568	−0.503–0.637
Spring SD	Spring standard deviation	0.02–0.304	0.028–0.279
Summer SD	Summer standard deviation	0.023–0.199	0.012–0.226

**Table 2 animals-14-03025-t002:** Average percentage of contribution/permutation importance of each variable for the historical and contemporary models. Values depicting the most important variables (>15) are signalled in bold. See Appendix A for further details on variable importance.

Variables	Historical	Contemporary
AnnTemp	0/0	0.004/0.049
TempWet	8.262/**24.348**	0.755/2.266
TempDri	**73.079**/**48.555**	**91.69**/**79.650**
PrecDri	0.153/0.136	1.377/6.594
PrecSeas	4.815/3.92	0.061/0.402
Autumn SD	3.421/4.641	0.174/0.163
Spring M	0.663/0.805	1.214/3.367
Spring SD	8.378/**15.229**	4.328/7.04
Summer SD	1.23/2.364	0.398/0.470

## Data Availability

Occurrence data are available from the authors upon request.

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
