# Peer review of "Temporal Range Dynamics of the Lataste’s Viper (Vipera latastei Boscá, 1878) in Doñana (Spain): Insights into Anthropogenically Driven Factors"

_animals, 2024, doi:10.3390/ani14203025_

Round 1

Reviewer 1 Report

Comments and Suggestions for Authors

The manuscript provides important insights on the ENM based distribution pattern assessment, future prediction of distribution and effect of climate change on a viper species from southern Spain. However, the manuscript needs some improvements and corrections before it can be accepted for publication. My comments, suggestions and corrections can be found on the attached pdf file.

Comments on the Quality of English Language

The language of the article needs some improvements, especially in some portions where it's difficult to understand what the authors wish to convey.

Author Response

REVIEWER #1

The manuscript provides important insights on the ENM based distribution pattern assessment, future prediction of distribution and effect of climate change on a viper species from southern Spain. However, the manuscript needs some improvements and corrections before it can be accepted for publication. My comments, suggestions and corrections can be found on the attached pdf file.

AUTHORS: Thank you very much for your revision. We have reviewed your comments in the PDF file and corrected the manuscript accordingly. Below you will find considerations on some of these comments.

Lines 3, 16 … - It should be Donana, Spain: Please correct throughout the MS

AUTHORS: the spelling “Doñana” is widely accepted, also in academic literature. Therefore, we prefer to keep it.

Line 65 - A couple of sentences regarding the reptilian diversity of Donana should be added.

AUTHORS: the scope of this paragraph is quite wide and consequently we added information about the diversity of some vertebrate groups.

Line 101 - it will be better to mention the elevation range of the species in Donana

AUTHORS: Valverde didn’t mention it, just referred to the habitats where the species was common. We have rephrased this sentence to better reflect this.

Reviewer 2 Report

Comments and Suggestions for Authors

The paper by Carmona-González et al. aimed to investigate the dynamics of the spatial and temporal distribution of Lataste's viper in the Doñana region and how these dynamics are affected by climate and landscape change. Using ecological niche modeling, they found that the range of Lataste's viper is shrinking in the Doñana region, and identified climate change as the main factor affecting the species' distribution. The findings highlight the species' vulnerability to climate change and call for targeted conservation measures.   I would like to recommend acceptance after minor editing of English language. Some minor suggestions are in the attached file.

Author Response

REVIEWER #2

The paper by Carmona-González et al. aimed to investigate the dynamics of the spatial and temporal distribution of Lataste's viper in the Doñana region and how these dynamics are affected by climate and landscape change. Using ecological niche modeling, they found that the range of Lataste's viper is shrinking in the Doñana region, and identified climate change as the main factor affecting the species' distribution. The findings highlight the species' vulnerability to climate change and call for targeted conservation measures.   I would like to recommend acceptance after minor editing of English language. Some minor suggestions are in the attached file.

AUTHORS: thank you very much for your revision. We have checked your comments in the pdf file and improved and corrected the manuscript accordingly.

Reviewer 3 Report

Comments and Suggestions for Authors

The paper is an interesting study of the range, ecological requirements, and changing distribution of a snake species that is difficult to study.  Using many different sources of data to compile the current and historical ranges of the species is a strength of this study.  The analyses seem well thought out and give interesting results.  As I note below, the Projected ranges need a lot more explanation.  Generally, the writing is OK but some of the English needs work and in some places of the manuscript it makes it hard to understand what the authors are trying to say.

One problem with this project is its scope.  It is a worthwhile effort to study the distribution and range contraction of the species in your particular area, which differs ecologically from other areas where these snakes live.  However, when you make conclusions about the species when only considering a population that is unusual ecologically (coastal) when most of the populations of the species live in different habitats, it is a problem.  You need to be clear that the environmental factors that are important for this coastal population may or may not be very important for most of the populations of the species.  I would be good if you could take a wider view in the Discussion to synthesize what is known about the species, not just your populations.  My more specific comments follow.

11. The writing is mostly fine but there are some problems with the English throughout, which sometimes makes it difficult to know what the authors are trying to say.

22. Lines 16, 23, 65. I don’t really know what you mean by “global reference”.  Do you mean hotspot?

33. L47. You state that climate change is the predominant factor.  No doubt it is very important, but most studies would say habitat destruction and degradation are currently more important than climate change.

44.  L105. I checked iNaturalist and there are quite a few records for the region.

55.  Introduction, and elsewhere.  The word “remarkable/remarkably” is used too much.

66.  L156. I don’t know what you are specifically referring to with “the water balance is generally deficient”.

77.  L180. Your records were gathered in 2022 but you are submitting this paper in 2024 and there are more iNaturalist records.

88. L270. You need to explain Projection models more.  I don’t understand.

99. L288-290. Delete.

110. Figures 3, 4. It is confusing the order you have these figures in.  You can’t compare historical to contemporary easily because they are not adjacent and I still don’t understand Projection.

111.  Figure 4. I find the terminology for underestimated and permanent to be misleading.  I don’t understand why the purple is called underestimated.  Permanent is not correct because further in the future the range could contract more.  You explain this a bit better in the text but it needs to be explained in the caption and better overall.

112.  L393-408. This an interesting contrast but it also highlights a potential problem with this study.  By only concentrating on a single, somewhat unusual area (as far as the ecology of the snakes are concerned), you do not have a complete understanding of the species.  You only studied a coastal area, so you found they occur in coastal areas.  The other study included a lot of mountains, so they found the snakes occurred in the mountains.  Sure, in your area, temperature was very important, but that doesn’t mean it is of high importance in the entire range of the species.

113.  L420. I think you are making a leap to say that the realized niche has shrunk over time.  I don’t see how your data really show that.  I am not familiar with the literature cited here so maybe I am missing something.  This needs to be explained much better or removed.

114.  L436. This statement that the temperatures that the snakes used previously are no longer available is an important result that should be emphasized.

115.  L491. I like that you have highlighted the most important areas for future research.

Overall, I recommend that this manuscript may be ready for publication after some revision.  I am willing to discuss my comments with the authors, if they would like.

Sincerely

Comments on the Quality of English Language

This is included in the main comments.

Author Response

REVIEWER #3

The paper is an interesting study of the range, ecological requirements, and changing distribution of a snake species that is difficult to study.  Using many different sources of data to compile the current and historical ranges of the species is a strength of this study.  The analyses seem well thought out and give interesting results.  As I note below, the Projected ranges need a lot more explanation.  Generally, the writing is OK but some of the English needs work and in some places of the manuscript it makes it hard to understand what the authors are trying to say.

One problem with this project is its scope.  It is a worthwhile effort to study the distribution and range contraction of the species in your particular area, which differs ecologically from other areas where these snakes live.  However, when you make conclusions about the species when only considering a population that is unusual ecologically (coastal) when most of the populations of the species live in different habitats, it is a problem.  You need to be clear that the environmental factors that are important for this coastal population may or may not be very important for most of the populations of the species.  I would be good if you could take a wider view in the Discussion to synthesize what is known about the species, not just your populations.  My more specific comments follow.

AUTHORS: thank you very much for your revision. In this new version we have addressed all your comments. We have explained what the “projected” range is, revised the English and worked on the discussion to better link our local study with other regional studies.

  1. The writing is mostly fine but there are some problems with the English throughout, which sometimes makes it difficult to know what the authors are trying to say.

AUTHORS: we have extensively revised the manuscript improving the English throughout

  1. Lines 16, 23, 65. I don’t really know what you mean by “global reference”.  Do you mean hotspot?

AUTHORS: Doñana is located within a biodiversity hotspot and this term is not appropriate to designate it. In this new version, we changed it to “a region of notable biodiversity richness”.

  1. L47. You state that climate change is the predominant factor.  No doubt it is very important, but most studies would say habitat destruction and degradation are currently more important than climate change.

AUTHORS: corrected

  1. L105. I checked iNaturalist and there are quite a few records for the region.

AUTHORS: we checked iNaturalist and there are just 6 records for our study area. We believe that these are “scarce contemporaneous records” as referred in the text and therefore we kept it as it was before.   

  1. Introduction, and elsewhere.  The word “remarkable/remarkably” is used too much.

AUTHORS: corrected

  1. L156. I don’t know what you are specifically referring to with “the water balance is generally deficient”.

AUTHORS: The water balance or budget is the relationship between water inputs and outputs. “Deficient” was now substituted by “negative”.

  1. L180. Your records were gathered in 2022 but you are submitting this paper in 2024 and there are more iNaturalist records.

AUTHORS: We consulted iNaturalist and there are two records gathered after September 2022. However, none of them are in areas where the species has not already been observed in our study. Therefore, we do not believe these additional records would significantly alter the results or conclusions of our study.

  1. L270. You need to explain Projection models more.  I don’t understand.

AUTHORS: Models can be transferred to other space and time scenarios. The algorithms apply the formula describing the species’ niche to another set of environmental variables, which can correspond to the past, to the future or to another geographical region or resolution scale in the present conditions. His procedure is called “projecting models” (Sillero et al. 2021).

In our study, besides the model predictions for each period, the historical model was projected to the contemporary period ('Projection'). This approach allowed us to predict the habitat suitability in contemporary times, considering the ecological relationships that were found in the historical models. This strategy has been used to study potential shifts in the realised niche shift between time periods. We have further explained this procedure in Material and Methods (lines 269-274).

Sillero, N.; Arenas-Castro, S.; Enriquez-Urzelai, U.; Gomes Vale, C.; Sousa-Guedes, D.; Martínez-Freiría, F.; Real, R.; Barbosa, A. M. Want to model a species niche? A step-by-step guideline on correlative ecological niche modelling. Ecol. Modell. 2021, 456, 109671. https://doi.org/10.1016/j.ecolmodel.2021.109671

  1. L288-290. Delete.

AUTHORS: deleted

  1. Figures 3, 4. It is confusing the order you have these figures in.  You can’t compare historical to contemporary easily because they are not adjacent and I still don’t understand Projection.

AUTHORS: because the “projection” is derived from historical models and reflect the contemporary period, we believe that this is the best way of representing model outputs. We expect that our explanation on what the “projection” is, help the reviewer to understand our way to make figure 3 and 4.

  1. Figure 4. I find the terminology for underestimated and permanent to be misleading.  I don’t understand why the purple is called underestimated.  Permanent is not correct because further in the future the range could contract more.  You explain this a bit better in the text but it needs to be explained in the caption and better overall.

AUTHORS: we have modified the text and the figure legend to better explain these terms.

In the text (lines 289-296), we now stated:

In the GIS, binary Historical, Contemporary and Projection were compared to quantify the temporal dynamics of available suitable areas in absence/presence of a potential shift in the realized niche. Consequently, we calculated the following categories: 1) lost, area identified in Historical only, representing the suitable areas that were lost in contemporary times; 2) underestimated, pixels identified in the Contemporary and Historical but not in Projection, referring to the pixels where the species could currently occur, reflecting a niche shift; 3) permanent, pixels identified in Historical, Projection and Contemporary, referring to the pixels where the species could currently occur, without considering a niche shift; 4) gained, pixels identified in the Contemporary only, representing pixels that became suitable in contemporary times, reflecting a niche shift.

In the figure legend, we rephrased to:

Lost habitats represent the suitable areas that were lost in contemporary times; permanent suitable habitats refer to pixels where the species could currently occur, without considering a niche shift; underestimated suitable habitats refer to pixels where the species could currently occur, mediated by a niche shift; and gained suitable habitats refer to pixels that became suitable in contemporary times, reflecting a niche shift.

  1. L393-408. This an interesting contrast but it also highlights a potential problem with this study.  By only concentrating on a single, somewhat unusual area (as far as the ecology of the snakes are concerned), you do not have a complete understanding of the species.  You only studied a coastal area, so you found they occur in coastal areas.  The other study included a lot of mountains, so they found the snakes occurred in the mountains.  Sure, in your area, temperature was very important, but that doesn’t mean it is of high importance in the entire range of the species.

AUTHORS: we agree with this comment and with the limitation that our study has in being carried out at the local scale of Doñana. It is not our intention to extrapolate our results to other regions. We have added some information in this regard (lines 398-400).

It must be noticed that, in this section, we are discussing our results in the light of other studies identifying the species ecological requirements, no matter of the scale and/or region. We found similarities in the ecological requirements, for instance, with north-central populations (Freitas et al., 2023; Scaramuzzi et al., 2023), as well as with some lineages within the species (Martínez-Freiría et al., 2021) or the whole species (Chamorro et al 2021). We found, however, what it seems an incongruence in relation to a previous study conducted at the Iberian scale (Santos et al 2006), discussing this apparent incongruence in a way of a similitude (i.e. mountain areas are the coldest areas within the species range, in a similar way to coastal areas).

Chamorro, D.; Martínez-Freiría, F.; Real, R.; Muñoz, A.-R. Understanding Parapatry: How Do Environment and Competitive Interactions Shape Iberian Vipers’ Distributions? Edited by D. Chapple. Journal of Biogeography 2021, 48, 1322–1335. https://doi.org/10.1111/jbi.14078.

Freitas, I.; Tarroso, P.; Zuazo, Ó.; Zaldívar, R.; Álvarez, J.; Meijide-Fuentes, M.; Martínez-Freiría, F. Local Niches Explain Coexistence in Environmentally-Distinct Contact Zones between Western Mediterranean Vipers. Scientific Reports 2023, 13, 21113.

Martínez-Freiría, F.; Freitas, I.; Zuffi, M.A.L.; Golay, P.; Ursenbacher, S.; Velo-Antón G. Climatic refugiaboosted allopatric diversification in Western Mediterraneanvipers. J Biogeogr. 2020,47:1698–1713. https://doi.org/10.1111/jbi.13861

Santos, X.; Brito, J.C.; Sillero, N; Pleguezuelos, J.M.; Llorente, G.A.; Fahd, S.; Parellada, X. Inferring Habitat-Suitability Areas with Ecological Modelling Techniques and GIS: A Contribution to Assess the Conservation Status of Vipera latastei. Biological Conservation 2006, 130 (3): 416–25. https://doi.org/10.1016/j.biocon.2006.01.003.

Scaramuzzi, A.; Freitas, I.; Sillero, N.; Martínez-Freiría, F. Meso-Habitat Distribution Patterns and Ecological Requirements of Two Mediterranean Vipers Depict Weak Competition in a Contact Zone. Journal of Zoology 2023, 320, 308–321. https://doi.org/10.1111/jzo.13087.

  1. L420. I think you are making a leap to say that the realized niche has shrunk over time.  I don’t see how your data really show that.  I am not familiar with the literature cited here so maybe I am missing something.  This needs to be explained much better or removed.

AUTHORS: we have rephrased this section to better describe the pattern that we found and to tone down our statements. Notice that we found concordant reductions in the niche breadth (three variables defined the species niche in historical times, while just one variable in current times) and in the potential suitable area of the species. In accordance to the literature (e.g. Scheele et al 2017), this pattern suggests that the species is declining in Doñana. 

Scheele, B.C.; Foster, C.N.; Banks, S.C.; Lindenmayer, D.B. Niche Contractions in Declining Species: Mechanisms and Con-sequences. Trends in Ecology & Evolution 2017, 32, 346–355. https://doi.org/10.1016/j.tree.2017.02.013.

  1. L436. This statement that the temperatures that the snakes used previously are no longer available is an important result that should be emphasized.

AUTHORS: in the conclusion (lines 495-496), we have emphasized that the major factor likely associated to the species range contraction is the increase of temperatures favoured by the ongoing climate change

  1. L491. I like that you have highlighted the most important areas for future research.

AUTHORS: thank you very much for your comments and help to improve our manuscript.

Overall, I recommend that this manuscript may be ready for publication after some revision.  I am willing to discuss my comments with the authors, if they would like.

Sincerely